Molecular phylogenies confirm the presence of two cryptic Hemimycale species in the Mediterranean and reveal the polyphyly of the genera Crella and Hemimycale (Demospongiae: Poecilosclerida)

Uriz Maria J. iosune@ceab.csic.es
Garate Leire
Agell Gemma
Department of Marine Ecology, Centre for Advanced Studies of Blanes (CEAB-CSIC) , Blanes, Girona , Spain
Toonen Robert
Electronic publication date: 2017 Mar 7
Publication date: 2017
Volume: 5
Electronic Location ID: e2958
Received 2016 Nov 19; Accepted 2017 Jan 4
Copyright: © 2017 Uriz et al.
Copyright year: 2017
Copyright holder: Uriz et al.
License: This is an open access article distributed under the terms of the Creative Commons Attribution License, which permits unrestricted use, distribution, reproduction and adaptation in any medium and for any purpose provided that it is properly attributed. For attribution, the original author(s), title, publication source (PeerJ) and either DOI or URL of the article must be cited.
License URL: https://creativecommons.org/licenses/by/4.0/

Keywords: Biodiversity, Molecular taxonomy, Morphological taxonomy, Cryptic species, Marine sponges, Phylogeny, Hemimycale mediterranea, Hemimycale columella, Hemimycale arabica, Crella cyathophora

Funding: MARSYMBIOMICS project Spanish MINECO, CTM2013-43287-P BluePharmTrain FP7 People-INT, Ref. 2013-667786 Grup Consolidat SGR-120 Benthomics project Spanish MICINN, CTM-2010-22218-C02-01 The research has been funded by MARSYMBIOMICS project (Spanish MINECO, CTM2013-43287-P), BluePharmTrain (FP7 People-INT, Ref. 2013-667786), and Grup Consolidat SGR-120, to Maria J. Uriz. Leire Garate benefited from a fellowship within the Benthomics project (Spanish MICINN, CTM-2010-22218-C02-01). The funders had no role in study design, data collection and analysis, decision to publish, or preparation of the manuscript.

==============================
Background

Sponges are particularly prone to hiding cryptic species as their paradigmatic plasticity often favors species phenotypic convergence as a result of adaptation to similar habitat conditions. Hemimycale is a sponge genus (Family Hymedesmiidae, Order Poecilosclerida) with four formally described species, from which only Hemimycale columella has been recorded in the Atlanto-Mediterranean basin, on shallow to 80 m deep bottoms. Contrasting biological features between shallow and deep individuals of Hemimycale columella suggested larger genetic differences than those expected between sponge populations. To assess whether shallow and deep populations indeed belong to different species, we performed a phylogenetic study of Hemimycale columella across the Mediterranean. We also included other Hemimycale and Crella species from the Red Sea, with the additional aim of clarifying the relationships of the genus Hemimycale.

Methods

Hemimycale columella was sampled across the Mediterranean, and Adriatic Seas. Hemimycale arabica and Crella cyathophora were collected from the Red Sea and Pacific. From two to three specimens per species and locality were extracted, amplified for Cytochrome C Oxidase I (COI) (M1–M6 partition), 18S rRNA, and 28S (D3–D5 partition) and sequenced. Sequences were aligned using Clustal W v.1.81. Phylogenetic trees were constructed under neighbor joining (NJ), Bayesian inference (BI), and maximum likelihood (ML) criteria as implemented in Geneious software 9.01. Moreover, spicules of the target species were observed through a Scanning Electron microscope.

Results

The several phylogenetic reconstructions retrieved both Crella and Hemimycale polyphyletic. Strong differences in COI sequences indicated that C. cyathophora from the Red Sea might belong in a different genus, closer to Hemimycale arabica than to the Atlanto-Mediterranean Crella spp. Molecular and external morphological differences between Hemimycale arabica and the Atlanto-Mediterranean Hemimycale also suggest that Hemimycale arabica fit in a separate genus. On the other hand, the Atlanto-Mediterranean Crellidae appeared in 18S and 28S phylogenies as a sister group of the Atlanto-Mediterranean Hemimycale. Moreover, what was known up to now as Hemimycale columella, is formed by two cryptic species with contrasting bathymetric distributions. Some small but consistent morphological differences allow species distinction.

Conclusions

A new family (Hemimycalidae) including the genus Hemimycale and the two purported new genera receiving C. cyathophora and Hemimycale arabica might be proposed according to our phylogenetic results. However, the inclusion of additional Operational Taxonomic Unit (OTUs) appears convenient before taking definite taxonomical decisions. A new cryptic species (Hemimycale mediterranea sp. nov.) is described. Morphologically undifferentiated species with contrasting biological traits, as those here reported, confirm that unidentified cryptic species may confound ecological studies.

Introduction

The discovery of cryptic species is continuously improving our knowledge on real ecosystem biodiversity and functioning, which are intimately related (Frainer, McKie & Malmqvist, 2014). Unrecognized cryptic diversity may mask biological features such as divergent reproduction patterns, growth dynamics, and inter-species interactions, among others (Knowlton, 1993; Prada et al., 2014; de Meester et al., 2016; Loreau, 2004), which may confound conservation studies (Forsman et al., 2010) and obscure the introduction pathway of invasive species (Knapp et al., 2015).

Molecular tools help to confirm suspected hidden species. However, molecular based identifications alone do not solve the problem of species misidentification, in particular when the cryptic species have overlapping distributions (e.g., Knowlton & Jackson, 1994; Tarjuelo et al., 2001; De Caralt et al., 2002; Blanquer & Uriz, 2007, 2008; Pérez-Portela et al., 2007). In these cases, deep studies on their morphology, biology (e.g., life-history traits), and ecology (e.g., growth dynamics) become crucial to understand the mechanisms underlying their coexistence (López-Legentil et al., 2005; Pérez-Portela et al., 2007; Blanquer, Uriz & Agell, 2008; Payo et al., 2013).

Sponges are sessile, aquatic filter-feeders that are widespread across oceans, depths, and ecosystems (Van Soest et al., 2012), with so far 8,789 accepted species inventoried in 2016 (Van Soest et al., 2016) and ca. 29,000 predicted to be discovered in the forthcoming years (Hooper & Lévi, 1994; Appeltans et al., 2012), many of which remain currently hidden among supposed widespread morpho-species (Uriz & Turon, 2012).

The poor dispersal capacities of sponges prevent in most cases gene flow among populations even at short geographical distances (Boury-Esnault, Pansini & Uriz, 1993; Uriz et al., 1998; Nichols & Barnes, 2005; Mariani et al., 2006; Uriz, Turon & Mariani, 2008). Consequently, sponge populations become genetically structured (Boury-Esnault, Pansini & Uriz, 1993; Duran, Pascual & Turon, 2004; Blanquer, Uriz & Caujapé-Castells, 2009; Guardiola, Frotscher & Uriz, 2012, 2016), which favors speciation, while the sponge plasticity fosters phenotypic (morphological) convergence to similar habitats (Blanquer & Uriz, 2008).

Many new cryptic sponge species have been discovered in the last decades thanks to the use of molecular markers (see Uriz & Turon, 2012 for a review until 2012, Knapp et al., 2011; de Paula et al., 2012). However, less often, molecularly discovered new species have also been described morphologically (but see Blanquer & Uriz, 2008; Cárdenas & Rapp, 2012; Reveillaud et al., 2011, 2012), which is necessary if phylogeny is aimed to translate into taxonomy, and the new species are wanted to be considered in ecological studies.

Sponge species can be both morphologically (e.g., Uriz & Turon, 2012) and, more rarely, molecularly (with the markers used) cryptic (Carella et al., 2016; Vargas et al., 2016) but show contrasting biological features. For instance, Scopalina blanensis (Blanquer & Uriz, 2008), which is sympatric with Scopalina lophyropoda, mainly grows in winter. Conversely, Scopalina lophyropoda regresses in winter and grows principally in summer–autumn (Blanquer, Uriz & Agell, 2008), thus indicating temporal niche partition.

The Order Poecilosclerida (Porifera: Demospongiae) harbors the highest number of species within the Class Demospongiae (Systema Porifera) and it is far from being resolved from a phylogenetic point of view (Morrow et al., 2012; Thacker et al., 2013). Within Poecilosclerida, the Family Hymedesmiidae represents a hotchpotch where genera of dubious adscription have been placed (Van Soest, 2002). As expected, this family appeared clearly polyphyletic in a molecular phylogeny of the so-called G4 clade based on 28S rRNA gene (Morrow et al., 2012).

Hymedesmiidae currently contains 10 accepted genera among which, Hemimycale Burton, 1934 (Van Soest et al., 2016). The position of genus Hemimycale, which shares with Hymedesmia, and Phorbas (Hymedesmiidae) and with Crella (Crellidae), the so-called aerolate areas with an inhaling function, has changed from Hymeniacidonidae in Halichondrida (Lévi, 1973) to Hymedesmiidae in Poecilosclerida (Van Soest, 2002). More recently, in 18S phylogenies of Poecilosclerida, Hemimycale columella was retrieved within the Crellidae clade, although with low support (Redmond et al., 2013).

Hemimycale harbors only four formally described species (Van Soest et al., 2016): the type species Hemimycale columella (Bowerbank, 1874), from Northwestern Atlantic and Mediterranean, Hemimycale rhodus (Hentchel, 1929) from the North Sea, Hemimycale arabica Illan et al., 2004 from the Red Sea and Hemimycale insularis Moraes, 2011 from Brazil. However, the simple spicule complement of the genus, which only consists of strongyles with some occasional styles, may propitiate the existence of morphologically (based on the spicules) cryptic species.

Hemimycale columella, the type species of Hemimycale, is widely distributed across the Atlanto-Mediterranean basin, from shallow (ca. 10 m) to deep (ca. 60 m) waters (Uriz, Rossell & Martín, 1992). Assays performed with eight microsatellite loci developed from deep specimens of Hemimycale columella (González-Ramos, Agell & Uriz, 2015) failed to amplify a high percentage of the assayed individuals from a shallow population, which suggested larger genetic differences than those expected between intra-species sponge populations.

Furthermore, the life cycle of species has been monitored in a shallow Northwestern Mediterranean population of what was thought to be Hemimycale columella (Pérez-Porro, González & Uriz, 2012), where all individuals disappeared after larval release in early November and new individuals arose the forthcoming year but on different rocky sites, which pointed to annual mortality and subsequent recruitment from sexually produced propagula (settling larvae). Conversely, during a study of deeper populations of Hemimycale columella (González-Ramos, Agell & Uriz, 2015), we recorded their survival for more than three years. Thus, shallow and deep populations of Hemimycale columella seemed to show contrasting life spans, which were thought to be a result of contrasting habitat characteristics. However, a 2-year monitoring of two, some km apart, populations (one deep and one shallow) and the main environmental factors at both locations, confirmed their contrasting life span and growth traits, as well as proved no correlation between biological features and environmental factors (M. J. Uriz, L. Garate & G. Agell, 2013–2014, unpublished data), which rather pointed to population intrinsic (genetic) differences.

To assess whether these two population types with contrasting biological traits but without clearly distinct morphological characters belonged or not to different species, we performed a phylogenetic study of individuals considered as Hemimycale columella across the Mediterranean, using three molecular (nuclear and mitochondrial) gene partitions. We incorporated additional species to the analyses to gain knowledge on the relationships between Hemimycale species and other genera of families Hymedesmiidae and Crellidae.

Materials and Methods

Sampling

Fragments of what a priori was thought to be Hemimycale columella were collected by SCUBA diving across the Northwestern, central and eastern Mediterranean, and Adriatic Sea, between 12 and 45 m of depth during several campaigns (Coconet, Benthomics, and MarSymbiOmics projects) (Table 1). Moreover, fragments of Hemimycale arabica and Crella cyathophora from the Red Sea (Dedalos and Ephistone) and Pacific (Bempton Islands) between 5 and 20 m depth were also collected (Table 1). Individuals were photographed underwater before sampling. Collected fragments were divided into two pieces, one of them was preserved in 100% ethanol, and after three alcohol changes, kept at −20 °C until DNA extraction; the other fragment was fixed in 5% formalin in seawater and preserved in 70% ethanol as a voucher for morphological and spicule studies. All vouchers have been deposited at the Sponge collection of the Centre d’Estudis Avançats de Blanes (numbers CEAB.POR.GEN.001 to CEAB.POR.GEN.029).

Table 1 Geographical origin and ecological distribution of the individuals used in the phylogenetic study, with accession numbers.

Species	Sea/Ocean	Locality	Voucher numbers	Accession numbers	
Hemimycale arabica ind. 1	Red Sea	Dedalos, Brother Islands	CEAB.POR.GEN.001	COI: KY002124
18S: KY002171
28S: KY002181	
Hemimycale arabica ind. 2	Red Sea	Elphinstone, Brother Islands	CEAB.POR.GEN.002	COI: KY002125
18S: KY002172
28S: KY002182	
Hemimycale columella	Northeastern Atlantic	Plymouth, Wales, UK		28S: HQ379300.1
18S: KC902127.1	
Hemimycale columella ind. 1	Northwestern Mediterranean	Arenys de Mar, Spain	CEAB.POR.GEN.003	28S: KY002183	
Hemimycale columella ind. 2	Northwestern Mediterranean	Arenys de Mar, Spain	CEAB.POR.GEN.004	28S: KY002184	
Hemimycale columella ind. 3	Northwestern Mediterranean	Arenys de Mar, Spain	CEAB.POR.GEN.005	COI: KY002126	
Hemimycale columella ind.1	Northwestern Mediterranean	Tossa de Mar, Spain	CEAB.POR.GEN.006	COI: KY002127
18S: KY002160
28S: KY002185	
Hemimycale columella ind. 2	Northwestern Mediterranean	Tossa de Mar, Spain	CEAB.POR.GEN.007	COI: KY002128
18S: KY002161
28S: KY002186	
Hemimycale columella ind. 3	Northwestern Mediterranean	Tossa de Mar, Spain	CEAB.POR.GEN.008	COI: KY002129
28S: KY002187	
Hemimycale columella ind. 4	Northwestern Mediterranean	Tossa de Mar, Spain	CEAB.POR.GEN.009	28S: KY002188	
Hemimycale mediterranea sp. nov. ind. 1	Northwestern Mediterranean	Tossa de Mar, Spain	CEAB.POR.GEN.010	COI: KY002130
18S: KY002162
28S: KY002189	
Hemimycale mediterranea sp. nov. ind. 2	Northwestern Mediterranean	Tossa de Mar, Spain	CEAB.POR.GEN.011	18S: KY002163
28S: KY002190	
Hemimycale mediterranea sp. nov. ind. 4	Northwestern Mediterranean	Tossa de Mar, Spain	CEAB.POR.GEN.012	COI: KY002131	
Hemimycale mediterranea sp. nov. ind. 5	Northwestern Mediterranean	Tossa de Mar, Spain	CEAB.POR.GEN.013	COI: KY002132	
H. mediterránea sp. nov. ind. 3	Adriatic Sea	Koznati, Croatia	CEAB.POR.GEN.014	COI: KY002134	
H. mediterránea sp. nov. ind. 7	Adriatic Sea	Koznati, Croatia	CEAB.POR.GEN.015	18S: KY002170
28S: KY002193	
H. mediterránea sp. nov. ind. 8	Adriatic Sea	Koznati, Croatia	CEAB.POR.GEN.016	28S: KY002194	
H. mediterránea sp. nov. ind. 2	Adriatic Sea	Tremity, Italy	CEAB.POR.GEN.017	COI: KY002133	
H. mediterránea sp. nov. ind. 11	Adriatic Sea	Tremity, Italy	CEAB.POR.GEN.018	28S: KY002199	
H. mediterránea sp. nov. ind. 8	Central Mediterranean	Porto Cesareo, Italy	CEAB.POR.GEN.019	18S: KY002164	
H. mediterránea sp. nov. ind. 9	Central Mediterranean	Porto Cesareo, Italy	CEAB.POR.GEN.020	18S: KY002165
28S: KY002197	
H. mediterránea sp. nov. ind. 10	Central Mediterranean	Porto Cesareo, Italy	CEAB.POR.GEN.021	28S: KY002198	
H. mediterránea nov. sp. ind. 5	Adriatic Sea	Karaburum, Albania	CEAB.POR.GEN.022	18S: KY002166
28S: KY002191	
H. mediterránea nov. sp. ind. 6	Adriatic Sea	Karaburum, Albania	CEAB.POR.GEN.023	18S: KY002167
28S: KY002192	
H. mediterránea sp. nov. ind. 3	Eastern Mediterranean	Othonoi, Greece	CEAB.POR.GEN.024	18S: KY002168
28S: KY002195	
H. mediterránea sp. nov. ind. 4	Eastern Mediterranean	Othonoi, Greece	CEAB.POR.GEN.025	18S: KY002169
28S: KY002196	
Crella cyatophora ind.1	Red Sea	Dedalos, Brother Islands	CEAB.POR.GEN.026	COI: KY002120
18S: KY002173
28S: KY002177	
Crella cyatophora ind. 2	Red Sea	Elphinstone, Brother Islands	CEAB.POR.GEN.027	COI: KY002121 18S: KY002174 28S: KY002178	
Crella cyatophora ind. 3	Pacific	Bempton Patch Reef (beween New Caledonian and Australia)	CEAB.POR.GEN.028	COI: KY002122
18S: KY002175
28S: KY002179	
Crella cyatophora ind. 4	Pacific	Bempton Patch Reef (between New Caledonian and Australia)	CEAB.POR.GEN.029	COI: KY002123
18S: KY002176
28S: KY002180	
Crella elegans	Mediterranean	France		18S: KC902282	
Crella elegans	Mediterranean	France		18S: AY348882	
Crella elegans	Mediterranean	France		28S: HQ393898	
Crella plana	Northeastern Atlantic	Northern Ireland		18S: KC9023009	
Crella rosea	Northeastern Atlantic	Northern Ireland		28S: HQ379299	
Crella rosea	Northeastern Atlantic	Northern Ireland		18S: KC902282	
Phorbas bihamiger	Northeastern Atlantic	English Channel		18S: KC901921.1
28S: KC869431	
Phorbas punctatus	Northeastern Atlantic	Wales		18S: KC869439.1 28S: KC869439.1	
Phorbas dives	Northeastern Atlantic	English Channel		28S: HQ379303	
Phorbas fictitioides	North Pacific	–		COI: HE611617.1	
Phorbas tenacior	Northeastern Atlantic	–		18S: AY348881	
Phorbas glaberrimus	Antarctic	Ross Sea		COI: LN850216.1	
Hymedesmia paupertas	Northeastern Atlantic			18S: KC902073.1
28S: KF018118.1	
Hymedesmia pansa				18S: KC902027.1	
Hymedesmia paupertas	Northeastern Atlantic			28S: KF018118.1	
Kirkpatrickia variolosa	Antarctic	Ross Sea		COI: LN850202.1	
Note:

Individuals sequenced de novo are in bold.

DNA extraction, amplification, and sequencing

DNA extractions were performed on two to three specimens per species and locality (totaling 18 individuals). Hemimycale spp. were extracted with QIAmp DNA stool kit (Qiagen), while Crella spp. were extracted with DNeasy Blood & Tissue kit (Qiagen) according to the manufacturer’s protocol. Standard primers were used for COI partitions M1–M6 (Folmer et al., 1994) and 18S rRNA (1F and 1795R, from Medlin et al., 1988), and Porifera primers for the D3–D5 partition of 28S rRNA (Por28S–830F and Por28S–1520R, from Morrow et al., 2012). Different amplification protocols were performed for each gene (Table 2). COI (M1–M6 partition) amplifications were performed in a 50 μL volume reaction, containing 37.6 μL H2O, 5 μL buffer KCl (BIORON; F Holzinger Sales & Support, Germany), 2 μL BSA, 2 μL dNTP (Sigma; Sigma_Aldrich, Germany), 1 μL of primers, 0.4 μL Taq (BIORON; F Holzinger Sales & Support, Germany), and 1 μL of genomic DNA. 18S rRNA amplifications were performed in a 50 μL volume reaction, containing 36.85 μL H2O, 5 μL buffer (Invitrogen, Carlsbad, CA, USA), 0.75 μL MgCl (Invitrogen, Carlsbad, CA, USA), 1.2 μL DMSO (dimethyl sulfoxide), 1 μL BSA, 1.5 μL dNTP (Sigma; Sigma_Aldrich, Germany), 1 μL of primers, 0.7 μL Taq (Invitrogen, Carlsbad, CA, USA), and 1 μL of genomic DNA. Finally, partition D3–D5 of 28S rRNA amplifications were performed in a 50 μL volume reaction, containing 36.85 μL H2O, 5 μL buffer (Invitrogen, Carlsbad, CA, USA), 0.75 μL MgCl (Invitrogen, Carlsbad, CA, USA), 2 μL BSA, 2 μL dNTP (Sigma; Sigma_Aldrich, Germany), 1 μL of primers, 0.4 μL Taq (Invitrogen, Carlsbad, CA, USA), and 1 μL of genomic DNA. Polymerase chain reaction products were purified and sequenced in both directions using Applied Biosystems 3730xl DNA analyzers in Macrogen, Korea.

Table 2 PCR conditions for the three partitions used (COI, 28S and 18S).

PCR Stage	COI (M1–M6)	28S (D3–D5)	18S	
Denaturalization	94 °C 2 min	94 °C 5 min	94 °C 5 min	
	35 cycles	35–40 cycles	30 cycles	
Denaturalization	94 °C 1 min	94 °C 1 min	94 °C 30 s	
Annealing	43 °C 1 min	50–55 °C 1 min	53 °C 30 s	
Elongation	72 °C 1 min	72 °C 1 min	72 °C 30 s	
Final elongation	72 °C 5 min	72 °C 5 min	72 °C 5 min	

Sequence alignment and phylogenetic reconstructions

Sequences of COI, 28S, and 18S were aligned using Clustal W v.1.81, once their poriferan origin was verified using BLAST (http://blast.ncbi.nlm.nih.gov/Blast.cgi), as implemented in Genieous 9.01 (Kearse et al., 2012). When sequences were identical, only one sequence per locality and species was included in the phylogenetic trees. After alignment, ambiguous regions were determined with Gblocks v.091 b software (Castresana, 2000), which removes from 1 to 4% of poorly aligned positions and divergent regions of an alignment of DNA. Representatives of family Hymedesmiidae (i.e., genera Phorbas and Hymedesmia) and Crambeidae (i.e., genera Crambe and Monanchora) were selected as outgroups. The inclusion of Crambeidae as an outgroup was decided because the species Hemimycale arabica had been reported to contain similar secondary metabolites (polycyclic guanidine alkaloids) to those of Crambe and Monanchora (Ilan et al. 2004).

JModelTest 0.1.1 (Posada, 2008) was used to determine the best-fitting evolutionary model for each dataset. The model GTR + I + G was used for both mitochondrial and nuclear genes. Phylogenetic trees were constructed under neighbor joining (NJ) (default parameters), Bayesian inference (BI), and maximum likelihood (ML) using Geneious software 9.01 (Kearse et al., 2012). NJ generates unrooted minimum evolution trees (Gascuel & Steel, 2006). BI analyses were performed with MrBayes 3.2.1 (Ronquist & Huelsenbeck, 2003). Four Markov Chains were run with one million generations sampled every 1,000 generations. The chains converged significantly and the average standard deviation of split frequencies was less than 0.01 at the end of the run. Early tree generations were discarded by default (25%) until the probabilities reached a stable plateau (burn-in) and the remaining trees were used to generate a 50% majority-rule consensus tree. ML analyses were executed with PhyML v3.0 program (Guindon & Gascuel, 2003; Guindon et al., 2005). The robustness of the tree clades was determined by a nonparametric bootstrap resampling with 1,000 replicates in PhyML. MrBayes and PhyML were downloaded by Genieous.

Incongruence length difference (ILD) test (PAUP 4.0b10) was run (Swofford, 2002) to verify sequence homogeneity or incongruence between the 18S rRNA and COI markers and the 18S and 28S rRNA markers. The ILD test indicated no significant conflict (p = 0.93 and p = 0.91, respectively) between the marker pairs to be concatenated. Thus, concatenated 18S COI and 18S–28S rRNA datasets were constructed for the species with sequences available for both markers. The phylogeny on the three genes concatenated was not performed due to the few species/individuals for which the three genes were available.

Phenotypic characters

To assess whether molecular differences among the target populations and species (Hemimycale columella, Senso latus, Hemimycale arabica, and C. cyathophora) were supported by morphological and spicule traits, the target species were observed both in situ and on recently collected samples. Moreover, spicules of all the species were observed through light and scanning electron microscopes (SEM) after removing the sponge organic matter from small (3 mm3) pieces of each individual by boiling them in 85% Nitric acid in a Pyrex tube and then washed three times with distilled water and dehydrated with ethanol 96% (three changes). A drop of a spicule suspension in ethanol per individual was placed on 5 mm diameter stuffs, air dry, and gold–palladium metalized (Uriz, Turon & Mariani, 2008) in a Sputtering Quorum Q150RS. Observation was performed through a Hitachi M-3000 Scanning Electron Microscope at the Centre d’Estudis Avançats de Blanes.

The electronic version of this article in Portable Document Format (PDF) will represent a published work according to the International Commission on Zoological Nomenclature (ICZN), and hence the new names contained in the electronic version are effectively published under that Code from the electronic edition alone. This published work and the nomenclatural acts it contains have been registered in ZooBank, the online registration system for the ICZN. The ZooBank LSIDs (Life Science Identifiers) can be resolved and the associated information viewed through any standard web browser by appending the LSID to the prefix http://zoobank.org/. The LSID for this publication is: urn:lsid:zoobank.org:pub:48910653-0343-4A8D-911F-3498A755F305. The online version of this work is archived and available from the following digital repositories: PeerJ, PubMed Central and CLOCKSS.

The electronic version of this article in Portable Document Format (PDF) will represent a published work according to the International Code of Nomenclature for algae, fungi, and plants, and hence the new names contained in the electronic version are effectively published under that Code from the electronic edition alone. In addition, new names contained in this work have been submitted to MycoBank from where they will be made available to the Global Names Index. The unique MycoBank number can be resolved and the associated information viewed through any standard web browser by appending the MycoBank number contained in this publication to the prefix “http://www.mycobank.org/mb/283905”. The online version of this work is archived and available from the following digital repositories: PeerJ, PubMed Central, and CLOCKSS.

Results

18S rRNA phylogeny

The resulting phylogeny using the 18S rRNA partition on 25 sequences (17 new) of 695 nt. (46 variable positions, from which 38 were parsimony informative) was congruent under BI, and ML and just differed in the position of Hemimycale arabica which appeared as a sister group of the remaining Crella spp. and Hemimycale spp. under NJ (Fig. S1). The representatives of the family Crambeidae (Monanchora) appeared as outgroups and the genus Phorbas was a sister group of the remaining species. In the BI, NJ, and ML trees, the genera Hemimycale and Crella appeared polyphyletic, with the Red Sea species Hemimycale arabica and C. cyathophora, far away from the Atlanto-Mediterranean Hemimycale and Crella species. The Atlanto-Mediterranean Crella formed a well-supported clade (1/81/98, posterior probability/bootstrapping values), which was the sister group of the Atlanto-Mediterranean Hemimycale (1/97/98). Moreover, the deep Hemimycale columella clustered with an Atlantic sequence downloaded from the GenBank (0.89/89/88) forming a separate clade from the also well-supported (1/97/98) group containing the shallow Mediterranean Hemimycale. No genetic differences for this partition were found among shallow individuals. In the BI and ML trees, the two individuals of Hemimycale arabica appeared in unresolved positions while they formed a poorly supported (75%) clade in the tree under the NJ criterion (not shown).

28S rRNA (D3–D5) phylogeny

The 28S rRNA (D3–D5) dataset comprised 31 sequences (24 new) of 623 nt. (84 variable positions from which, 60 parsimony informative).

The resulting phylogenies were congruent with the three clustering criteria and matched in most cases the phylogeny based on the 18S rRNA partition, although the supporting values of some clades were in some cases slightly lower (Fig. S2).

The three phylogenies retrieved the representatives of Family Crambeidae (Monanchora and Crambe) as an outgroup. The monophyly of the in-group containing Crella spp. and Hemimycale spp. was strongly supported under the BI, NJ, and ML criteria (1/100/100). The genus Phorbas was a sister group of the remaining species considered. Crella was polyphyletic, with C. cyathophora separated from the well-supported clade (1/100/100) encompassing the Atlanto-Mediterranean Crella. The latter appeared as a sister clade of a poorly supported group (0.7/77/70) harboring C. cyathophora and Hemimycale spp. The Hemimycale spp. group, although monophyletic, was poorly supported under the NJ and ML criteria (77/70) while the Atlanto-Mediterranean Hemimycale clade was well supported under the three clustering criteria (1/92/95).

The deep and shallow Mediterranean populations of Hemimycale formed two well-supported monophyletic groups (0.96/87/83 and 0.96/ 100/98, for deep and shallow individuals, respectively), the former containing the Atlantic sequence of Hemimycale columella. No genetic differences for this partition were retrieved for shallow individuals despite their spread distribution across the Mediterranean. The individuals of C. cyathophora from the Red Sea clustered with those from the Pacific collected between Australia and Nouvelle Caledonie (1/89/76).

COI phylogeny

The COI dataset included 21 sequences (15 new) of 535 nt. (169 variable positions, from which 149 parsimony informative).

The COI phylogeny, which was congruent under BI, NJ, and ML, also retrieved the representatives of Crambeidae as outgroups of the group formed by Crella, Phorbas, and Hemimycale. The genus Phorbas clustered with the Atlanto-Mediterranean Crella spp. (0.98/100/86) likely because we only included one individual/species of Phorbas (Fig. S3).

A clade containing Hemimycale spp. and C. cyathophora was well supported (0.94/94/80). The Hemimycale clade was divided into two subclades corresponding to deep and shallow individuals. No genetic differences among shallow individuals were found. A sister, well supported group (1/100/94) contained C. cyathophora and Hemimycale arabica representatives with almost no genetic differences between them (Fig. S3).

Concatenated trees

The concatenated 18S + 28S rRNA (Fig. 1) confirmed the outgroup position for the Crambeidae representative (Monanchora), the polyphyly of Crella with the Red Sea and Pacific species forming a separate clade (1/100/100) from the Atlanto-Mediterran Crella, which appeared in a non-resolved position. Hemimycale also appeared polyphyletic, but the position of Hemimycale arabica was unresolved. The Atlanto-Mediterranean Hemimycale clade was confirmed as well as its division into two subclades: one containing the deep Mediterranean individuals together with two Atlantic sequences of the species and the other one harboring the shallow Mediterranean individuals, which did not show any genetic difference across the Mediterranean and Adriatic Sea.

Figure 1 Phylogenetic tree using concatenated (18S rRNA + COI) partitions.

BI, NJ and ML gave the same topologies. Posterior probability, neighbor joining, and maximum likelihood supporting values are at the base of clades.

The concatenated 18S rRNA + COI (Fig. 2) tree contained only 13 sequences and no representative of Crambeidae could be included. The representatives of the Atlanto-Mediterranean Crella appeared as outgroups of the remaining target species, which formed two well-supported clades: one containing C. cyathophora and Hemimycale arabica representatives (1/100/100) and the other with the Atlanto-Mediterranean Hemimycale (1/100/100) divided into two monophyletic well-supported groups (deep and shallow individuals).

Figure 2 Phylogenetic tree using concatenated (18S + 28S rRNA) partitions.

BI, NJ and ML gave the same topologies. Posterior probability, neighbor joining, and maximum likelihood supporting values are at the base of clades.

Discussion

The phylogenetic reconstructions performed with 18S, 28S rRNA and COI, as well as with concatenated genes (18S rRNA + COI and 18S + 28S rRNA) support the polyphyly of Crella and Hemimycale, under the three clustering criteria used. As although Hemimycale was monophyletic with the 28S rRNA (D3–D5) marker, the clade was not statistically supported.

Crella cyathophora sequences differ from those of the Atlanto-Mediterranean Crella spp. in 2% (18S rRNA), 2.19% (28S rRNA), and 10.24% (COI). These genetic distances suggest that, despite some spicule similitude (presence of acanthoxeas and smooth diactines with Atlanto-Mediterranean Crella spp.), the former species belongs in a different genus, closer to Hemimycale arabica (0.71% with 18S rRNA, 1.37% with 28S rRNA, and none with COI) than to the Atlanto-Mediterranean Crella spp.

Hemimycale arabica differs from the Atlanto-Mediterranean Hemimycale spp. in 1.43–1.86% with 18S rRNA, 1.78–2.19 with 28S rRNA, and in 8.38–8.64% with COI. These strong COI differences and the contrasting morphological traits (blue external color, irregular, rim-free, aerolate areas and abundance of true styles in Hemimycale arabica vs. orange–pinkish color, circular, rimmed aerolate areas, and slightly asymmetrical any strongyles almost exclusively in Hemimycale spp.) also indicate that Hemimycale arabica would belong in a different genus, which might also include C. cyathophora, as there are not COI differences between these two species.

Moreover, the Atlanto-Mediterranean Crellidae appeared in 18S and 28S rRNA phylogenies as a sister group of the Atlanto-Mediterranean Hemimycale, which suggests higher affinities of this genus with Crellidae than with Hymedesmiidae (its current family). However, more complete analyses including additional Crellidae and Hymedesmiidae OUT’s are needed to move Hemimycale from Hymedesmiidae to Crellidae.

The phylogenetic trees with any of the three gene partitions either separately or concatenated confirm the presence of two cryptic Hemimycale species in the Mediterranean within what was considered until now Hemimycale columella. The new species that we name Hemimycale mediterranean sp. nov. (see description below) has a shallower distribution across the whole Mediterranean than Hemimycale columella, which has Atlantic affinities. Hemimycale columella differs from Hemimycale mediterranea in 0.85% (18S rRNA), 1.23% (28S rRNA), and in 1–1.2% (COI).

The lack of genetic diversity among the distant populations of Hemimycale mediterranea analyzed points to its recent presence in the Mediterranean, which is compatible with a recent introduction. However, the new species has not been recorded out of the Mediterranean, and thus, its origin cannot be established at the present time.

Many cryptic species that were revealed by molecular markers have never been formally described owing to the difficulty of finding diagnostic phenotypic characters. Although after exhaustive observation, only slight, morphological differences have been found to differentiate Hemimycale mediterranea sp. nov. from Hemimycale columella (see species description below), these phenotypic differences are consistent across individuals and thus, add to molecular differences and biological traits (L. Garate et al., 2013–2014, unpublished data) to consistently differentiate these two species.

Species description

Genus Hemimycale Burton, 1934

Sequence accession Numbers GenBank (Table 1)

Type species Hemimycale columella (Bowerbank, 1874)

Hemimycale is the only genus of Hymedesmiidae that has smooth diactines and monactines exclusively (Van Soest, 2002). The genus was described by Burton (1934) as “reduced Mycaleae with skeleton of loose fibers of styli, sometimes modified into anisostrongyles, running vertically to the surface; fibers tending to branch and anastomose; no special dermal skeleton, no microscleres.”

The spicule complement described by Burton; however, seems different from that reported in the several modern redescriptions of Hemimycale columella (Vacelet, Donadey & Froget, 1987), which report predominant straight anisostrongyles with rare or absent styles. Indeed, Burton stated that the Bowerbank representation of Hemimycale columella spicules was wrong because it figured anisostrongyles instead of styles, and was precisely the dominance of styles what induced Burton to place the species among the Mycaleae. The termination of the diactines either round or pointed ends may be the result of different silica concentration in the water masses, as reported for other siliceous sponge skeletons (Uriz, 2006), but it cannot be totally discarded that the Burton Hemimycale columella belonged in another Hemimycale species.

Species: Hemimycale columella (Bowerbank, 1874)

Sequence accession numbers GenBank (Table 1)

Description (Figs. 3A–3D): Encrusting to massive sponges. Surface smooth, covered with circular inhaling, areas up to 6 mm in diameter with an up to 3 mm high rim. Morbid and fleshy consistence. Translucent to whitish ectosome, difficult to separate from the choanosome. Thousands of calcareous spherules, 1 μm in diameter formed by intracellular calcifying bacteria (Uriz et al., 2012) are spread through the sponge mesohyl and specially accumulated at the sponge periphery of whitish individuals (L. Garate et al., 2013–2014, unpublished data).

Figure 3 In situ pictures of Atlanto-Mediterranean Hemimycale spp.

(A, B, C, D) Hemimycale columella from 35 to 40 m of depth. (E, F, G, H) Hemimycale mediterranea sp. nov. from 12 to 17 m of depth. Whitish tinge is due to calcibacteria accumulation. Red tinges are due to several species of epibiotic cyanobacteria. Arrows point to aerolate inhaling areas; arrowheads indicate the epibiont cyanophycea on Hemimycale mediterranea specimens.

Color from pinkish-orange to whitish outside, dark orange inside.

Spicules (Table 3; Fig. 4F): Asymmetric strongyles (anysotrongyles), straight, 302–435 × 3–4 μm in size. Styles rare or completely absent from the Mediterranean specimens (this study) and Canary Islands (Cruz, 2002).

Table 3 Locality and spicule sizes of the studied individuals, and comparison with descriptions by other authors.

Species	Author	Locality	Depth (m)/Assemblage	Styles	Strongyles (range/mean)	Acanthoxeas	
Hemimycale arabica ind. 1	This study	Red Sea (Egypt)	14/coral reef	160–189 (179.6) × 7–8 (7.5)	210–290 (273) × 2.8–4.1 (3.6)	–	
Hemimycale arabica	Illan et al. 2004	Red Sea (Egypt)		190–250 (218) × 3.5–5 (4.7)	200–290 (266) × 2.5–4 (3.5)	–	
H. mediterránea ind. 7	This study	Adriatic (Croatia)	10–15/rocky sub-horizontal	–	233–330 (274.8) × 3–4.6 (4.0)	–	
H. mediterránea ind. 11	This study	Adriatic (Italy)	10–15/rocky sub-horizontal	–	251–300 (276.6) × 2.1–4 (3.0)	–	
H. mediterránea ind. 5	This study	Adriatic (Albania)	10–15/rocky sub-horizontal	–	274–317 (296.4) × 2.9–4.5 (4.0)–	–	
H. mediterránea ind. 10	This study	Central Med. (Italy)	10–15/rocky sub-horizontal	–	229–328 (291.3) × 2.4–5.2 (3.5)	–	
H. mediterránea ind. 3	This study	Eastern Med. (Greece)	10–15/rocky sub-horizontal	–	242–340 (272.7) × 2.6–4 (3.2)	–	
H. mediterránea ind. 1	This study	NW Med. (Spain)	12–16/rocky wall	–	261–320 (296.3) × 3.1–3.8 (3.5)	–	
Hemimycale columella ind. 1	This study	NW Med. (Spain)	27–29/coralligenous	–	302–435 (370) × 3–4 (3.7)	–	
“Hemimycale columella”	Vacelet 1987	NW Med. (France)	–	–	225–310 (285) × 2–4 (3)	–	
Hemimycale columella	Vacelet 1987	NW Med. (France)	–	–	320–410 (369) × 2.5–3.8 (3.1)	–	
“Hemimycale columella”	Vacelet 1987	NW Med. (France)	–	–	220–320 (273) × 2–4 (2,7)	–	
Hemimycale columella	Vacelet 1987	North Atlantic (France)	–	–	290–465 (394) × 4–7 (5.1)	–	
Hemimycale columella	Topsent 1925	North Atlantic (France)	–	–	400 × 6	–	
“Hemimycale columella”	Foster 1995	North Atlantic (UK)	–	–	330–420 (373) × 5–6 (5.85)	–	
Hemimycale columella	Bowerbank 1874	North Atlantic (UK)	–	–	376 × 7	–	
Crella cyatophora ind. 3	This study	Indo-Pacific (Bemptom)	18m/coral reef	–	205–308 (263.9) × 2.2–4.3 (3.4)	92–115 (105.4) × 2–2.3 (2)	
C. cyatophora ind. 1	This study	Red Sea (Egypt)	12/coral reef	–	227–293 (267.8) × 2.5–3.9 (3.4)	89–120 (109.4) × 1.8–2.5(2.47)	

Figure 4 Spicules of Hemimycale spp. and Crella cyathophora though SEM.

(A, B, C, D, E) Anysostrongyles (Hemimycale mediterranea). (F) Anisostrongyles (Hemimycale columella). (G) Anisostrongyles and one style (Hemimycale arabica). (H) Anysotrongyles and acantoxeas (Crella cyatophora). Inserts on each picture correspond to magnifications of the spicule ends.

Skeletal arrangement: Plumose branching bundles of anysostrongyles together with spread spicules. A palisade of vertical anysotrongyles forms the rim around the inhaling areas.

Distribution: Northeastern Atlantic (United Kingdom and Ireland coasts) Canarias Islands (Cruz, 2002), western Mediterranean: Tossa de Mar, Arenys de Mar, from 28 to 60 m depth (this study). It is not possible to confirm whether previous Mediterranean records of the species (see Vacelet & Donadey, 1977) belonged to Hemimycale columella or to Hemimycale mediterranea.

Biology: Multiannual life span, ca. 60% survival after two monitoring years; maximum growth in autumn–winter (L. Garate et al., 2013–2014, unpublished data). Larval release occurs at the beginning of November in Mediterranean populations (M. J. Uriz, L. Garate & G. Agell, 2013–2014, unpublished data).

Species: Hemimycale mediterranea sp. nov. (Figs. 3E–3H)

Sequence accession numbers GenBank (Table 1)

Description: Thick encrusting sponges with aerolate inhaling areas up to 3 mm in diameter, surrounded by an up to 1.5–2 mm high rim, which in some cases barely surpasses the sponge surface. Thousands of calcareous spherules, 1 μm in diameter formed by intracellular calcifying bacteria are spread through the sponge mesohyl and specially accumulated at the sponge periphery (Garate et al., in press).

Ectosome: Firmly attached to the choanosome.

Color: Flesh to clear brownish externally, more or less whitish depending on calcibacteria accumulation at the surface, sometimes partially covered by an epibiotic (reddish or pinkish) cyanobacteria.

Spicules (Table 3; Figs. 4A–4E): Smooth, uniform in size, straight, anysostrongyles, 200–296 × 3–4 μm in size. Styles completely absent.

Skeletal arrangement: Plumose undulating bundles of anysostrongyles together with spread spicules. A palisade of vertical anysotrongyles forms the rim around the inhaling areas.

Known distribution: Northwestern Mediterranean, central Mediterranean, Adriatic, eastern Mediterranean (Spain: Cap De Creus, Tossa, Blanes, Arenys, South Italy: Croatia, Tremiti, Turkey, Greece) between 3 and 17 m deep.

Biology: Annual life span, maximum growth rates in summer (M. J. Uriz, L. Garate & G. Agell, 2013–2014, unpublished data). Larval release at the end of September beginning of October (M. J. Uriz, L. Garate & G. Agell, 2013–2014, unpublished data).

In most cases, it is difficult to ascertain whether individuals of Hemimycale columella recorded by other authors belong to Hemimycale columella or Hemimycale mediterranea. The redescription of Hemimycale columella by Van Soest (2002) based on the holotype (from the Atlantic) reported large aerolate porefields with elevated rims, which are shared with the deep Mediterranean specimens of Hemimycale columella (Figs. 3A–3D) in contrast to the small, short-rimmed porefields showed by Hemimycale mediterranea sp. nov. Both species have mainly straight slightly asymmetric strongyles but the spicule sizes are systematically larger in Hemimycale columella (Table 3). However, while styles were rarely present in Hemimycale columella individuals, they have not been found in specimens of Hemimycale mediterranea sp. nov. The external color also differs between the two species, being orange to pinkish in Hemimycale columella and flesh color to brownish Hemimycale mediterranea sp. nov. (Figs. 3E–3H). Vacelet & Donadey (1977) reported two different color forms occurring side by side on the littoral of Provence (France), one pink cream and the other one brownish. Likely the second color morph, which besides had smaller strongyles, corresponded to the Hemimycale mediterranea sp. nov.

Color has not received much attention as a diagnostic character in sponges because it has been generally considered to be a response to higher or lower light irradiance at the sponge habitat, or to the presence of epibiotic or symbiotic cyanobacteria. However, color has proven to be taxonomically relevant to distinguish other invertebrates such as shrimp species (Knowlton & Mills, 1992) and also sponge species of the genus Scopalina (Blanquer & Uriz, 2008), and thus it seems worthy to be taken into account in sponge taxonomy.

The slight phenotypic differences found between the two species appear; however, consistent across individuals and localities within the Atlanto-Mediterranean basin. Moreover, their ecological distribution and bacterial symbionts, strongly differentiate these two cryptic species. For instance, although calcareous spherules produced by intracellular bacteria are present in the two species, the producer bacteria belong in different species (Garate et al., in press), and the respective microbial communities totally differ (Garate et al., in press). Symbionts, as predators do (e.g., Wulff, 2006), often distinguish their target sponge preys or hosts while the species remain morphologically cryptic to taxonomists. Moreover, Hemimycale mediterranea sp. nov. shows an annual life span, with individuals disappearing after larval release, while Hemimycale columella has a multiannual life span (M. J. Uriz, L. Garate & G. Agell, 2013–2014, unpublished data) and growth dynamics also differs between the two species, as Hemimycale mediterranea sp. nov. grows more in summer, while Hemimycale columella grows preferentially in autumn–winter (M. J. Uriz, L. Garate & G. Agell, 2013–2014, unpublished data).

The contrasting ecological distribution of these two cryptic species in the Mediterranean helps to predict their identity in the field. Hemimycale mediterranean sp. nov. inhabits shallower zones than Hemimycale columella. However, it is likely that both species may share occasionally habitats, as the record of the two color morphs side by side (Vacelet & Donadey, 1977) indicate. Hemimycale mediterranea sp. nov. seems to be more abundant and widespread in the Mediterranean than Hemimycale columella. Molecular differences between groups of individuals of Hemimycale columella suggest the possible presence of additional cryptic species among the deep Mediterranean Hemimycale.

The presence of two morphologically cryptic Hemimycale species in the Mediterranean, which show contrasting biological traits, reinforces the idea that cryptic species represent something more than wrong taxonomic identifications or biodiversity underestimates. They may feature contrasting biological cycles and life spans, and puzzle biological studies, which may invalidate conservation policies based on those studies.

Supplemental Information

Supplemental Information 1 New sequences used in the study.

18S, 28S, and COI partitions used in the phylogenetic study and deposited at GenBank.

Click here for additional data file.

Supplemental Information 2 Phylogenetic tree using the 18S rRNA partition.

BI, NJ and ML gave almost the same topologies. The two individuals of H. arabica that appeared in unresolved positions under BI and ML formed a poorly supported (75%) clade in the tree under the NJ criterion (not shown). Posterior probability, neighbor joining and maximum likelihood supporting values are at the base of clades.

Click here for additional data file.

Supplemental Information 3 Phylogenetic tree using the 28S rRNA (D3–D5 partition).

BI, NJ and ML gave congruent topologies. Posterior probability, neighbor joining and maximum likelihood supporting values are at the base of clades.

Click here for additional data file.

Supplemental Information 4 Phylogenetic tree using the COI (M1–M6 partition).

BI, NJ and ML gave the same topologies. Posterior probability, neighbor joining and maximum likelihood supporting values are at the base of clades.

Click here for additional data file.

We thank J. Sureda, M. Bolivar, and F. Crespo for sampling support in the Mediterranean Sea. M. Bolivar also provided some deep Hemimycale pictures. X. Turon and E. Ballesteros provided samples of Hemimycale arabica from the Read Sea and C. cyathophora from Bempton Islands (Pacific), respectively.

Additional Information and Declarations

Competing Interests

Author Contributions

DNA Deposition

Data Deposition

New Species Registration

The authors declare that they have no competing interests.

Maria J. Uriz conceived and designed the experiments, analyzed the data, contributed reagents/materials/analysis tools, wrote the paper, and reviewed drafts of the paper.

Leire Garate conceived and designed the experiments, performed the experiments, analyzed the data, wrote the paper, prepared figures and/or tables, and reviewed drafts of the paper.

Gemma Agell performed the experiments, analyzed the data.

The following information was supplied regarding the deposition of DNA sequences:

The 18 S, 28 S and COI sequences described here are accessible at GenBank https://www.ncbi.nlm.nih.gov/genbank/.

Accession numbers: KY002124, KY002171, KY002181, KY002125, KY002172, KY002182, KY002183, KY002184, KY002126.

COI: KY002127.

18S: KY002160, KY002185, KY002128, KY002161, KY002186, KY002129, KY002187,KY002188, KY002130, KY002162, KY002189, KY002163, KY002190, KY002131, KY002132, KY002134, KY002170, KY002193, KY002194, KY002133, KY002199, KY002164, KY002165, KY002197, KY002198, KY002166, KY002191, KY002167, KY002192, KY002168, KY002195, KY002169, KY002196, KY002120, KY002173, KY002177, KY002121, KY002174, KY002178, KY002122, KY002175, KY002179, KY002123, KY002176, KY002180.

The following information was supplied regarding data availability:

CEAB Porifera Collection.

Vouchers: CEAB.POR.GEN.001, CEAB.POR.GEN.002, CEAB.POR.GEN.003, CEAB.POR.GEN.004, CEAB.POR.GEN.005, CEAB.POR.GEN.006, CEAB.POR.GEN.007, CEAB.POR.GEN.008, CEAB.POR.GEN.009, CEAB.POR.GEN.010, CEAB.POR.GEN.011, CEAB.POR.GEN.012, CEAB.POR.GEN.013, CEAB.POR.GEN.014, CEAB.POR.GEN.015, CEAB.POR.GEN.016, CEAB.POR.GEN.017, CEAB.POR.GEN.018, CEAB.POR.GEN.019, CEAB.POR.GEN.020, CEAB.POR.GEN.021, CEAB.POR.GEN.022, CEAB.POR.GEN.023, CEAB.POR.GEN.024, CEAB.POR.GEN.025, CEAB.POR.GEN.026, CEAB.POR.GEN.027, CEAB.POR.GEN.028, CEAB.POR.GEN.029.

The following information was supplied regarding the registration of a newly described species:

Publication LSID: urn:lsid:zoobank.org:pub:69255188-5A55-4D5C-9DC2-43E2B6CF6997.

Species name: LSID: urn:lsid:zoobank.org:act:A90CB361-6EC5-4B94-805B-F5A4C0EDF2F9.

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
