# Peer review of "Molecular phylogenies confirm the presence of two cryptic Hemimycale species in the Mediterranean and reveal the polyphyly of the genera Crella and Hemimycale (Demospongiae: Poecilosclerida)"

_PeerJ, doi:10.7717/peerj.2958_

## Round 0.1 · original submission · Major Revisions

We now have comments back from 2 expert referees who both imply that the work should be publishable, but requires major revisions prior to being acceptable for publication. While both referees are enthusiastic about the work, each also has the same major concerns about the lack of clear morphological analyses included in this submission, and the impression that the single gene analyses add little other than length to the primary story. I agree with the referees that some morphological data would be exceedingly useful in this regard - whether it supports the molecular analyses or not, it is important to include and will add to the growing body of literature that evaluates how (and if) the common morphological traits used for sponge taxonomy support or contradict the molecular analyses being performed. Likewise, while I believe the single gene analyses are useful and valuable to provide, I see their point, and as such these analyses might be better relegated to the online supplementary materials. I also agree with the referees on some additional suggestions for improvement including that without data or a reference, statements asserting differences in the microbial community among the species should be removed, and that better attention to a rather extensive existing literature with less focus on self-citations is probably warranted. Beyond these major comments, there are numerous minor comments both in the reviews and in the attached mark-up manuscript returned by the referees. I expect most of these comments will be relatively straightforward to deal with, but will take some major revision to the submission, so I look forward to seeing your revised manuscript.

Reviewer 1 ·

Basic reporting

Clear, unambiguous, professional English language used throughout.
Reviewer’s response: There were very few grammatical errors throughout the manuscript. The manuscript was well written. I have addressed these as “sticky notes” on the Urizpeerj-reviewing-14535-v0_minor comments.pdf.

Intro & background to show context.
Reviewer’s response: The introduction gives good background on the importance of using molecular markers for the discovery of cryptic species. The authors also give good informative background on the taxonomy of the Family Hymedesmiidae as well as the natural history of Hemimycale spp.

Literature well referenced & relevant.
Reviewer’s response: Yes, although some references supporting the author’s assumption that state that 0-1.2% sequence differences is enough to differentiate species should be included in the discussion.

Structure conforms to PeerJ standards, discipline norm, or improved for clarity.
Reviewer’s response: (Yes)

Figures are relevant, high quality, well labelled & described.
Reviewer’s response: The phylogenetic trees were well presented. However, the color images need to have subheadings for their panels (i.e. Figure 7A, B, C, D). I have some minor suggested corrections for the authors to consider attached as “sticky notes” in the highlighted Urizpeerj-reviewing-14535-v0_minor comments.pdf.

I also think the authors are redundant in providing in vivo photographic images of H. columella in Figure 6 and of H. mediterranea sp. nov. in Figure 8. I recommend combining Figures 6 and 8 into one single figure. The authors should also consider highlighting what they want the reader to look at when referring to a figure panel. Each panel of the figure should be referred to in the main body of the paper.

Can the authors provide a scale bar for photographic images underwater in Figure 6 and Figure 8?

Can the authors provide the accession numbers in Figures 1-3 for the sequences that the authors submitted?

In Figure 7, what is the tiny micrograph overlapping the images to the left and right of it? It makes the figure confusing to follow. I would suggest removing it or incorporating it into the figure if it is a main point in the paper.

Raw data supplied (see PeerJ policy).
Reviewer’s response: The authors provide voucher catalogue numbers and accession numbers of all submitted sequences for this paper. The metadata associated with the sample collections were also included.

Experimental design

Original primary research within Scope of the journal.
Reviewer’s response: Yes

Research question well defined, relevant & meaningful. It is stated how the research fills an identified knowledge gap.
Reviewer’s response: The manuscript illustrates well defined research questions that fills an identified knowledge gap.

Rigorous investigation performed to a high technical & ethical standard.
Reviewer’s response: Authors present a robust phylogenetic analysis to answer their research questions.

The paper does lack morphological data or discussion points to support the author’s argument that Crella cyathophora should be considered a Hemimycale. Can the authors discuss some morphological data that would further support this argument?

Throughout the abstract (lines 34-35) the authors mention that “external morphological differences between H. arabica and the Atlanto-Mediterranean Hemimycale spp. are different enough to belong to a separate genus” but they never address what these morphological differences are in the discussion. I recommend that the authors include more information on how these sponges differ morphologically to one another in the discussion.

Methods described with sufficient detail & information to replicate.
Reviewer’s response: (Yes)

Validity of the findings

Data is robust, statistically sound, & controlled.
Reviewer’s response: Yes although some morphological data should have been included (see general comments section below).

Conclusions are well stated, linked to original research question & limited to supporting results.

Reviewer’s response: The discussion does a good job of supporting the validity of the findings from a molecular perspective but does not address morphological data to either support or reject the molecular data (see comments below).

Additional comments

I enjoyed reading the manuscript prepared by Uriz, Garate and Agell. Their key findings were: 1. Sponges within the genus Crella and Hemimycale are polyphyletic. 2. Nuclear and mitochondrial phylogeny showed that Crella cyathophora should belong to the genus Hemimyclae. 3. Hemimycale arabica might belong to a different genus. 4. Species description of Hemimycale mediterranea sp. nov. based on molecular, life history and morphological data and 5. propose the new family (Hemimycalidae).

The authors did a good job in presenting their data using a robust phylogenetic analysis of their findings. I have highlighted my recommended minor changes as “sticky notes” on the “urizpeerj-reviewing-14535-v0.pdf” and a list of “major comments” to be addressed below. Some of the highlighted text in the “Urizpeerj-reviewing-14535-v0_minorcomments.pdf” are for my personal notes. Please disregard any highlighted text that are not supplemented with a “sticky note”.

My major comments to the authors regarding each of their key findings:

4a. The authors showed strong support for the polyphyletic nature of Crella cyathophora and Hemimycale arabica. I agree with the authors that Crella cyathophora also seems to be more closely related to the genus Hemimycale and more specifically to H. arabica from a molecular standpoint. However, there a few points that need to be addressed. For example, the authors did not provide any discussion points on the morphological traits that C. cyathophora and Hemimycale share that would strengthen this hypothesis provided by the molecular data. Is there a way that the authors can provide either some morphological evidence that would hold true for their genetic results? If not, could the authors provide some more discussion points describing the molecular similarities provided by previous taxonomic work on both species that would help describe how these sponges are related. I can see that the authors also provide some SEM spicule images of C. cyathophora in Figure 7 but never discuss how these might be similar or different to sponges in the Hemimycale genus.

4b. The authors should include a statement in the discussion about the original description of Hemimycale arabica and how it might be distinguishable from other Hemimycale spp. based on morphology. Should it belong to a different genus based on morphological differences as well?

4c. In the species description of the discussion, the authors do a good job of comparing the morphological differences between H. columnella and H. mediterranea sp. nov. (paragraph starting at line 372). However, I am not sure if the molecular evidence (lines 298-299) “ H. mediterranea in 0.85% (18S), 1.23% (28S), and in 1-1.2% (COI)” is enough to make the assumption that these are cryptic species. Previous studies publishing cryptic species based on molecular data show >2% sequence dissimilarity between cryptic species. The authors did show strong bootstrap values for the separation of these two species, but I am not sure that their dissimilarities (0-1.2%) are convincing enough? I recommend that the authors discuss and cite other phylogenetic analyses from cryptic species that would support their argument.

4d. (paragraph starting at 372-384) The authors mention very important morphological characters that are helpful to the diagnosis between H. columnella. Then in lines 392-396 the authors continue to discuss three additional characters from unpublished work (Garate et al., unpublished data). For the purpose of this publication I think the authors provide enough morphological evidence to distinguish both species in the field. I am not sure that mentioning how the microbial communities differ between the two species are appropriate to discuss in this paper as no data are present. The authors should either remove this statement or should specifically say which bacterial groups are present and add this data to a supplementary file.

4e. For proper taxonomic documentation of the new species the authors should provide sponge tissue sections of the new species Hemimycale denoting tangential sections of the ectosome and cross sections that include the ectosome and choanosome. The skeletal arrangement from these sections could also provide important details of their morphology that would aid in the proper classification of either Hymimycale spp.

4f. I agree with the author’s in that they provide preliminary evidence that their proposed new family (Hemimycalidae) should be considered in the future as more species are resolved for this family.

Annotated reviews are not available for download in order to protect the identity of reviewers who chose to remain anonymous.

Reviewer 2 ·

Basic reporting

Uriz and colleages presented two cryptic Hemimycale species from the Mediterranean by using three different markers and documented the polyphyly of the genera Crella and Hemimycale.

This study is well written (minimal typographic/language errors, see comments) and presents important findings on these two genera which are worth to publish after changes are done.

The illustrations are partly of good quality (see e.g. comments for Illustration on Fig. 7). The Materials and Methods section for PCR reactions are recommended to be reduced. Concerning the phylogenetic reconstruction methods used, several major changes are suggested to be done (for details see comments below) E.g. it is unclear for the reader why single gene analyses are performed, as in the main answers to the research questions are clear from the concatenated analyses. Furthermore in the M&M part only concatenated datasets are mentioned. There are several improvements on tree-topology comparison and other detailed proposed changes concerning the phylogenetic reconstruction (see comments below for more details).


- Line 66: I would recommend using the currently valid number of accepted species from 2016 (8,789) not as inventoried at the World Porifera Database.

- Line 71: As there are many other publications “dispersal capacities of sponges please add e.g. in front of your listed references (Uriz et al., 1998 ect.) as there are other publications demonstrating this for various species like e.g.
Mariani, S., Uriz, M., Turon, X., and Alcoverro, T. (2006). Dispersal strategies in sponge
larvae: Integrating the life history of larvae and the hydrologic component. Oecologia 149, 174–184.
Nichols, S. A., and Barnes, P. A. G. (2005). A molecular phylogeny and historical
biogeography of the marine sponge genus Placospongia (Phylum Porifera) indicate low
dispersal capabilities and widespread crypsis. Journal of Experimental Marine Biology and Ecology 323, 1–15
Boury-Esnault, N., Pansini, M., and Uriz, M. J. (1993). Cosmopolitism in sponges: The
“complex” Guitarra fimbriata with description of a new species of Guitarra from the
northeast Atlantic. Scientia Marina 57, 367–373.

- Line 72-73: The statement that sponge populations are genetically structured was also made in the following publication, which is missing in the reference list:
Boury-Esnault, N., Pansini, M., and Uriz, M. J. (1993). Cosmopolitism in sponges: The
“complex” Guitarra fimbriata with description of a new species of Guitarra from the
northeast Atlantic. Scientia Marina 57, 367–373.
Therefore, please add all references or put “e.g.” in front of the reference list, so that it is clear for the reader that there are also other publications dealing with the above mentioned statement.

- Line 94: Please add a reference for the then accepted genera of the family Hymedesmiidae (e.g. Van Soest, WPD)

- Line 153-162: This paragraph can be reduced by only mentioning the differences of reagents used for the three genes.

- Line 326: please add a reference for statement: “as reported for other siliceous sponge skeletons”.

- Line: 379: Grammar correction: “they have not been found”

General: In my opinion all the single gene analyses (including figures) can be placed in the supplementary material as the main answer to the research questions and conclusion of the paper: which is the confirmation of the presence of two cryptic Hmimycale species in the Mediterranean as well as the polyphyly of both genera (Crella and Hemimycale) is already presented by the concatenated analyses. If there are other important reasons why the single gene analyses are worth including in the main MS it must be explained in more detail.

General comment to Fig. 7: The SEM pictures of all Hemimycale spp. species are of poor quality, thus key diagnostic characters not visible. I would highly recommend redoing the SEM and generating more zoomed-in pictures of the tips of the anysostrongyles and acanthoxeas in order to better verify the species. This would also improve and highlight the difficulties of using only morphology in taxonomy and new species descriptions. Please also correct the in the caption of the figure acanthoxas by adding an “e” acanthoxeas.

General comment to Fig. 8 and Figure 6: I would suggest to better indicating in the pictures the observed morphological features. This can be done by adding an arrows to morphological observed features or letters to the pictures e.g. point in Fig. 6 to the observed circular inhaling areas ect. and explain a bit more what the different in situ pictures present (maybe include this in the caption of the Figures).

Experimental design

- Line 101-104: Please keep the same reference style for the species names. Some are in parenthesis some not! Please also keep the shortcut for the genus names consistent. Sometimes you write Hemimycale sometimes only H. throughout the text!

- Line 105-106: You mean diactine strongyles and monactine styles? I would suggest removing the pharanthesis for the second style name.

- Line 114: Please indicate here for which species of Hemimycale the life cycle has been monitored. Was it also for H. columella?

- Line 153-162: This paragraph can be reduced by only mentioning the differences of reagents used for the three genes.

- Line 174-176: Here you mention that the Phylogenetic trees were constructed under NJ, BI and ML criteria in Geneious and below you indicate that mention the other programs PhyML ect. Here it is unclear, of weather you used MrBayes and PhyML as implemented tools in Geneious or not. Please make it more clear for the reader.

- Line 185-187: Why not doing the same test for different marker combinations e.g. CO1-28S? Please explain in more detail.

- Line 188-191: You also performed single gene analyses (see your Figures 1-3). Please add this information and explain.

- Line 211: Please explain why you choose Crambeidae as an outgroup? Was this family already successfully used in another study?

- Line 326: please add a reference for statement: “as reported for other siliceous sponge skeletons”.

- General: NJ issue: The authors used the Neighbour Joining (NJ) methods, which is considered as a phenetic one, and interpret them the same way like ML and BI phylogenetic methods. It should be made clearer that the ancestry, thus the relationships in NJ methods are never considered. This should be considered while interpreting the data and corrected, changed in the M&M part as well as in the Result part.

- General: Please indicate for each final alignment used (18S, 28S, CO1), how many of the characters were phylogenetically informative/uninvormative or contant. This can be done easily by loading the alignments in Paup.

- General: you mention for all three reconstructed single gene analyses that the phylogeny/topology was “mostly congruent under BI, NJ, and ML”. Please indicate where there were differences. I suggest doing a tree topology test to significantly demonstrate the congruencies/differences for each analysis.

Validity of the findings

- Line 297: please add sp. nov. to the new mentioned species of H. mediterranean
- Line 314: please add the sequence accession numbers or refer to Table 1 like you did for H. columella in line 330.
- Line 361: here you mention that the tissue is sometimes partially covered by reddish or pinkish cyanobacteria. Was there any investigation that it is really a epibiontic cyanobacteria? Or is it only a suggestion, so maybe the reason is this cyanobacterium? Please make this clearer to the reader! If you know the specific cyanobacterium it might be worth to mention it here!
- Line 365: what do you mean by spread spicules? other loose anysostrongyles or different spicules? Please make it clearer.
- Line 359: “r” is missing: “more or less”.
- Line 351: the author shows spicules of H. mediterranea in Figure 7 (see also my comment for this Fig. 7), which should be included here as well, additionally to the mentioned Fig. 8.

Additional comments

General: In my opinion all the single gene analyses (including figures) can be placed in the supplementary material as the main answer to the research questions and conclusion of the paper: The confirmation of the presence of two cryptic Hmimycale species in the Mediterranean as well as the polyphyly of both genera (Crella and Hemimycale) is already presented by the concatenated analyses. If there are other important reasons why the single gene analyses are worth including in the main MS it must be explained in more detail.

---

## Round 0.2 · accepted · Accept

I have now heard back from the more critical of the referees on the original submission, and both they and I are satisfied with the revisions to your manuscript in response to the initial round of reviews. GIven that, I am happy to move your manuscript into production at this time.

Reviewer 1 ·

Basic reporting

.

Experimental design

.

Validity of the findings

.

Additional comments

I am satisfied with the changes the reviewers have incorporated to their manuscript.